# Prediction of Increased Intracranial Pressure in Traumatic Brain Injury Using Quantitative Electroencephalogram in a Porcine Experimental Model

**DOI:** 10.3390/diagnostics13030386

**Published:** 2023-01-20

**Authors:** Ki-Hong Kim, Heejin Kim, Kyoung-Jun Song, Sang-Do Shin, Hee-Chan Kim, Hyouk-Jae Lim, Yoonjic Kim, Hyun-Jeong Kang, Ki-Jeong Hong

**Affiliations:** 1Department of Emergency Medicine, Seoul National University Hospital, Seoul 03080, Republic of Korea; 2Laboratory of Emergency Medical Services, Seoul National University Hospital Biomedical Research Institute, Seoul 03080, Republic of Korea; 3Clinical Trials Center, Seoul National University Hospital, Seoul 03080, Republic of Korea; 4Department of Emergency Medicine, Seoul National University Boramae Medical Center, Seoul 07061, Republic of Korea; 5Department of Emergency Medicine, Seoul National University College of Medicine, Seoul 03080, Republic of Korea; 6Department of Biomedical Engineering, Seoul National University College of Medicine, Seoul 03080, Republic of Korea

**Keywords:** traumatic brain injury, intracranial pressure, electroencephalogram, prediction model, machine learning

## Abstract

Continuous and non-invasive measurement of intracranial pressure (ICP) in traumatic brain injury (TBI) is important to recognize increased ICP (IICP), which can reduce treatment delays. The purpose of this study was to develop an electroencephalogram (EEG)-based prediction model for IICP in a porcine TBI model. Thirty swine were anaesthetized and underwent IICP by inflating a Foley catheter in the intracranial space. Single-channel EEG data were collected every 6 min in 10 mmHg increments in the ICP from baseline to 50 mmHg. We developed EEG-based models to predict the IICP (equal or over 25 mmHg) using four algorithms: logistic regression (LR), naive Bayes (NB), support vector machine (SVM), and random forest (RF). We assessed the performance of each model based on the accuracy, sensitivity, specificity, and AUC values. The accuracy of each prediction model for IICP was 0.773 for SVM, 0.749 for NB, 0.746 for RF, and 0.706 for LR. The AUC of each model was 0.860 for SVM, 0.824 for NB, 0.802 for RF, and 0.748 for LR. We developed a machine learning prediction model for IICP using single-channel EEG signals in a swine TBI experimental model. The SVM model showed good predictive power with the highest AUC value.

## 1. Introduction

Traumatic brain injury (TBI) is a growing concern with a huge contribution to mortality and disability worldwide [1]. Each year, 69 million victims are estimated to suffer from TBI, with disparity according to resources [2]. Severe TBI has a long-lasting effect and is associated with caregiver burden in multiple dimensions [3,4,5]. Prompt assessment and management with effective neuromonitoring modalities are needed to improve clinical outcomes.

Assessment of increased intracranial pressure (IICP) due to post-injury hyperemia or structural lesions is essential for evaluating the severity and preventing secondary injury in TBI [6]. Current guidelines recommend reducing intracranial pressure (ICP) based on mental status or radiographic examination to maintain perfusion in the brain tissue [7]. Since early decompression leads to favorable neurological outcomes for severe TBI [8] many efforts have been made to assess IICP and reduce the time gap of treatment [9,10,11].

A recent study revealed that ICP monitoring is essential to determine the appropriate time for therapeutic intervention and improve clinical outcomes [12,13]. However, measuring ICP requires invasive procedure such as inserting the pressure catheter into the intracranial cavity, which carries a considerable risk of complications [14]. Several non-invasive techniques have been applied to measure ICP. Transcranial Doppler has been demonstrated as a potential screening tool [15,16] and several studies have focused on monitoring the optic nerve sheath diameter to detect IICP [17,18]. However, a correlation was not found for each modality in a recent study [19].

Electroencephalography (EEG) has been used in qualitative and quantitative analyses to evaluate brain function. It can represent the cellular viability of brain tissue by electrical activity through non-invasive methods [20]. In a previous study, the EEG index calculated using power spectrum analysis correlated with ICP [21]. Studies conducted in an intensive care unit have also demonstrated a potential relationship between EEG activity and ICP changes [22,23].

Measuring ICP in TBI is important to recognize IICP, and developing a modality to measure ICP continuously and non-invasively could be useful for treating TBI. We hypothesized that a meaningful prediction model for IICP could be retrieved by analyzing EEG signals in TBI patients. A machine learning algorithm was used to develop a prediction model for IICP in recent studies and showed good discrimination ability [24,25]. The purpose of this study was to develop and validate an electroencephalogram (EEG)-based machine-learning prediction model for IICP in a swine TBI experimental model.

## 2. Materials and Methods

### 2.1. Ethical Statement

The animal experimental protocol was approved by the Institutional Animal Care and Use Committee of the study institution (IACUC number: 19-0097-S1A0 and 20-0115-S1A0). All animal care procedures complied with the Laboratory Animal Act of the Korean Ministry of Food and Drug Safety.

### 2.2. Study Design and Setting

An experimental swine TBI model was designed to manipulate ICP levels and acquire spontaneous EEG and hemodynamic data, including ICP data. The EEG was retrieved from the subject under diverse levels of ICP in the supine position. The experimental process consisted of three phases: the baseline, injury, and recovery phases. At the baseline phase, physiological data and blood gas analyses were acquired after all surgical preparations. During the injury phase, the balloon of the Foley catheter placed in the subdural space was inflated into the target range of ICP in steps (10 mmHg level from 20 mmHg to 50 mmHg, each 6 min period) to implement the TBI situation. After reaching an ICP of >50 mmHg, the Foley catheter balloon was deflated in the reverse order in the recovery phase. Throughout the experimental course, single-channel EEG and hemodynamic data, including ICP, were measured uninterruptedly. The test scenario is illustrated in Figure 1.

### 2.3. Experimental Animal and Housing

Thirty farm-raised crossbred female pigs aged approximately 3 months (41.3 ± 2.9 kg) were enrolled. The animals were maintained in an accredited Association for Assessment and Accreditation of Laboratory Animal Care (AAALAC) International (#001169) facility, in accordance with the Guide for the Care and Use of Laboratory Animals. They were considered healthy and were fasted overnight.

### 2.4. Surgical Procedure and Haemodynamic Measurements

The animals were initially sedated with intramuscular injections of 2–4 mg/kg tiletamine hydrochloride and zolazepam hydrochloride (Zoletil, Virbac, France) and 2 mg/kg xylazine (Rompun, BayerKorea, Seoul, Republic of Korea). They were orally intubated with an endotracheal tube connected to a capnograph (CapStar-100; CWE Inc., Ardmore, PA, USA). Isoflurane at a dose of 2–5% was injected to anaesthetize the animals, and mechanical ventilation was initiated. To continue anesthesia, a tidal volume of 10–12 mL/kg, respiratory rate of 14–18 breaths/min, partial pressure of arterial carbon dioxide at approximately 30–50 mmHg, and partial pressure of arterial oxygen over 80 mmHg were used. After creating a burr hole in the cranium, a pressure catheter (Mikro-Tip pressure catheter, Millar Instruments, Houston, TX, USA) was inserted to measure the ICP. The appropriate depth of the catheter was determined by lowering it until a consistent respiratory waveform was observed. An additional burr hole was perforated for insertion of a Foley catheter to manage the ICP level. Another Mikro-Tip pressure catheter was inserted into the femoral artery to measure arterial blood pressure (ABP). Electrocardiograms (ECG) were recorded to calculate heart rates. All data were gathered and recorded using a data acquisition platform (PowerLab 16/35, AD Instruments, Dunedin, New Zealand) at a rate of 1 kHz.

### 2.5. EEG Signal Acquisition

Battery-powered single-channel digital electroencephalography and disposable electrodes (MT-100, Kendall, ON, Canada) were installed to measure the cutaneous EEG under a referential montage. The device was designed to be securely attached to the rounded forehead. The reference and ground electrodes were attached to either side of the mastoid. An active electrode underneath the device was secured to the forehead. A schematic diagram and a picture of the installation of the EEG and surgical preparation on the head of the subject are shown in Figure 2. The raw EEG signal was band-pass filtered with a frequency band within 0.5–47 Hz and amplified with a gain of 12,000 *V*/*V*. The processed signal was digitized using a high-resolution analogue-to-digital converter (ADS1292, Texas Instruments, Dallas, TX, USA) and transmitted to a laptop via Bluetooth communication at a rate of 250 Hz. The computer application software received and saved EEG data.

### 2.6. Data Processing

Raw EEG was segmented into 2-sec long epochs with 1.5-sec overlaps with the next epoch. EEG indices in the time, frequency, and entropy domains were calculated in 0.5 s increments. ICP, heart rate (HR), and mean arterial pressure (MAP) were synchronized with EEG parameters at the same time intervals. Through the whole EEG indexes, the top 10 representative EEG parameters were selected using neighborhood component analysis (NCA) and Student’s t-test. The EEG parameters were normalized using the min-max normalization method, and NCA was applied to determine the feature ranking. The Student’s t-test was then applied to evaluate the discrimination performance of each parameter. The 10 parameters with the highest ranking and *p*-value below 0.05 were finally adopted for the prediction model development. The flow of data processing and EEG index selection is described in Appendix A
Figure A1 and Table A1. 

### 2.7. Prediction Model Development and Validation

A prediction model for IICP using EEG parameters and machine learning algorithms was developed and validated. IICP was defined as ICP levels equal to or greater than 25 mmHg, according to a pre-specified threshold [26]. We divided the subjects into a derivation group (70%, N = 21) and a validation group (30%, N = 9). Four machine learning methods were used to develop the prediction models: Logistic regression (LR), Naïve bayes (NB), Support Vector Machine (SVM), and Random Forest (RF). For LR, a generalized logistic model using binomial distribution was used. In the NB model, a kernel-smoothing method for the Gaussian function was applied. The SVM model adopted a Gaussian kernel function whose kernel scale and box constraint were 1.7 and 1, respectively. Finally, an ensemble aggregation method with a random subspace was used in the RF model. The number of ensemble learning cycles is 30. For all the models, 5-fold cross validated classifiers were created.

### 2.8. Statistical Analysis

Descriptive analysis calculating the median and interquartile range for each EEG index using IICP was conducted for each subject. Statistical significance was considered at a *p*-value < 0.05. To evaluate the performance of the prediction models, a confusion matrix with accuracy, sensitivity, and specificity was derived with 95% confidence intervals (CIs). An additional area under the receiver operating characteristic curve (AUC) were obtained. Data analyses were performed using MATLAB (MATLAB 2020a, The Mathworks, Natick, MA, USA) and SPSS (IBM SPSS Statistics 25, IBM SPSS Statistics, New York, NY, USA).

## 3. Results

A total of 30 pigs were tested. From each subject, baseline data for 5 min and 4-step injury phase for 24 min were obtained. Considering the stabilization period for hemodynamic parameters during ICP uprising, the first 1-minute data of each step were excluded. Finally, 25 min of data from each pig, near 612 min from all the pigs (10,410 data from baseline phase, and 63,041 data from injury phase), were analyzed. Using the collected data, 41% were determined as IICP (n = 25,438) with ICP equal to or greater than 25 mmHg. The first 21 pigs were enrolled in the derivation group and the latter 9 for the validation group. The experiments were conducted according to a predetermined protocol. Table 1 shows the baseline characteristics of the participants included in the analysis. There was no statistically significant difference in the arterial blood gas analysis results between the groups; however, physiological parameters, including ICP, were different between the two groups. 

The tendencies of the EEG indices were compared according to ICP ranges. All 10 EEG parameters showed statistically significant differences (Table 2). Among the 10 EEG indices, the magnitude of EEG, Log energy entropy, SD alpha, SD beta, and SD gamma increased according to the ICP range elevation by 10 mmHg. Two indices (DELTAR and DTABR) showed U-shape and inverse U-shaped characteristics. In injury phase, DELTAR and GAMMAPR increased gradually while DTABR and THETAPR decreased. It shows that the proportion of the higher frequency components (equal to or greater than 13 Hz) increased when an increased ICP was applied.

The prediction performance of each machine learning model for the IICP in the validation set is presented in Table 3. The accuracy of IICP differentiation was 0.773 (95% CI 0.694–0.853) for the SVM algorithm, 0.749 (95% CI 0.658–0.840) for the NB algorithm, 0.746 (95% CI, 0.673–0.818) for the RF algorithm, and 0.706 (95% CI 0.625–0.787) for the LR algorithm. The AUC of IICP differentiation was 0.860 (95% CI, 0.781–0.938) for the SVM algorithm, 0.824 (95% CI 0.740–0.907) for the NB algorithm, 0.802 (95% CI, 0.710–0.894) for the RF algorithm, and 0.748 (95% CI 0.644–0.851) for the LR algorithm. (Figure 3) All models showed good discrimination power for predicting IICP using EEG signals in TBI patients. Among all the models, the SVM algorithms showed higher accuracy than the other algorithms. Confusion matrices of validation data for each prediction models were demonstrated in Figure 4.

## 4. Discussion

In this study, EEG-based IICP prediction models using a machine-learning algorithm were developed and validated using a porcine TBI model. Based on EEG parameters acquired from single-channel EEG signals, four machine learning methods were used to predict IICP in TBI non-invasively. The performance of the prediction model showed good discriminatory ability, with an AUC > 0.8. Among the four models, the SVM method showed higher performance than the other methods. This experimental study suggests that the EEG-based IICP prediction model developed in this study could be implemented to assess ICP in a non-invasive manner during the management for TBI patients with TBI.

ICP monitoring in TBI is important for preventing secondary injuries and improving clinical outcomes. Bedside invasive procedures with cranial trephination and intracranial catheter insertion are conventionally performed to directly measure ICP. It causes serious complications such as brain tissue injury, cerebral hemorrhage, and infection. It is difficult to implement in the prehospital phase or in early trauma resuscitation in the emergency department. If predicting IICP in a non-invasive manner is available, a clinical decision to transfer patients to neurosurgical facilities or early intervention to reduce ICP could be conducted.

Non-invasive ICP prediction is usually established based on transcranial Doppler (TCD). The pulsatility index, which is a quantitative and qualitative analysis of the TCD morphology, has been widely used [27,28]. Otherwise, cerebral perfusion pressure estimation combined with arterial blood pressure has been widely studied [29]. However, TCD methods have not shown sufficient performance for application in the field. It also requires a skilled examiner to find the correct vessels and interpret the measurements, which can inevitably cause variation among medical providers and limitations for application [30].

EEG indices have been used to evaluate the prognostic value of brain injury in cardiac arrest [31,32]. A recent study showed that, based on machine learning techniques, EEG indices can predict cerebral perfusion pressure in cardiac arrest experimental studies with high discriminative power [33]. The advantage of the IICP prediction model based on EEG indexes is that it is objective and independent of the experience of medical personnel. EEG-based IICP prediction models using only portable EEG devices can be operated automatically without complex manipulation or interpretation processes.

In our study, the EEG characteristics showed a meaningful tendency according to the ICP range. We hypothesized that cerebral electrical activity changes according to ICP affect the characteristics and distribution of the EEG signals. The selected EEG indices were found to have linear, U-shaped, or reverse U-shaped tendencies. When reaching a certain level of ICP, it was expected that the pattern would change because of the decrease in cell viability due to damage. In addition, EEG flattening was remarkable during the recovery phase. Cerebral electrical activity could not properly reflect the ICP change in the recovery phase after severe cerebral damage [34].

Further research should focus on the adjustment of the model derived from this study to human subjects. Patterns of EEG indices would differ from our study, and a clinical study with a sufficient sample size would be required. Additionally, beyond predicting IICP, an advanced EEG-based prediction model should be developed that can estimate a more detailed range of ICP to evaluate not only IICP but also the relief of ICP elevation. Other non-invasive physiological parameters, such as heart rate or pulse oximetry, could be combined with the model to improve the prediction performance.

### Study Limitations

This study had several limitations. First, it was a preliminary animal experimental study with a limited sample size. Therefore, it is difficult to generalize our results to patients with TBI. Data acquisition from human subjects under different ICP conditions and feature analyses must be performed. Second, EEG signals cannot be obtained during the recovery phase. As described above, serious brain damage (especially ICP > 40 mmHg) most likely suppresses neuronal electrical activity. Third, this was a retrospective study. To perform real-time monitoring, technical improvements in the EEG device, signal processing techniques, and machine learning algorithms should be achieved. Lastly, there might be uncorrected artefact signals that could influence the EEG signals, even when we removed all identifiable artefacts and other stimuli during the experiment.

## 5. Conclusions

In a porcine experimental study using a traumatic brain injury model, the EEG-based IICP prediction model using machine learning techniques showed good discriminatory performance with an AUC of 0.86. This result suggests the potential usefulness of non-invasive EEG measurements for predicting IICP during early resuscitation and emergency care for TBI patients.

## Figures and Tables

**Figure 1 diagnostics-13-00386-f001:**
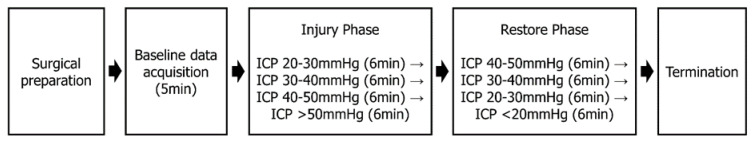
Experimental protocol.

**Figure 2 diagnostics-13-00386-f002:**
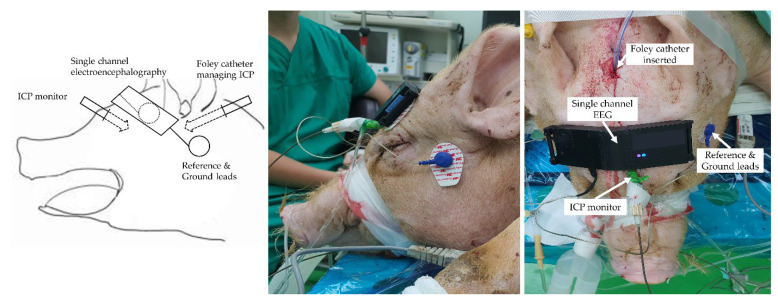
Attachment of single channel EEG in a porcine TBI experimental model. ICP, Intracranial pressure; EEG, Electroencephalogram; TBI, Traumatic brain injury.

**Figure 3 diagnostics-13-00386-f003:**
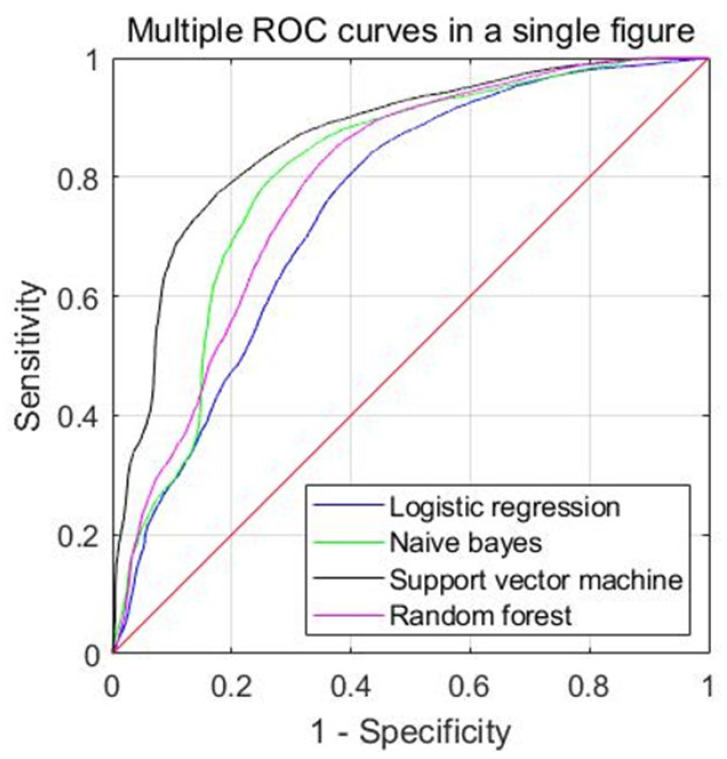
AUC value of each EEG based machine-learning prediction model for IICP in TBI.

**Figure 4 diagnostics-13-00386-f004:**
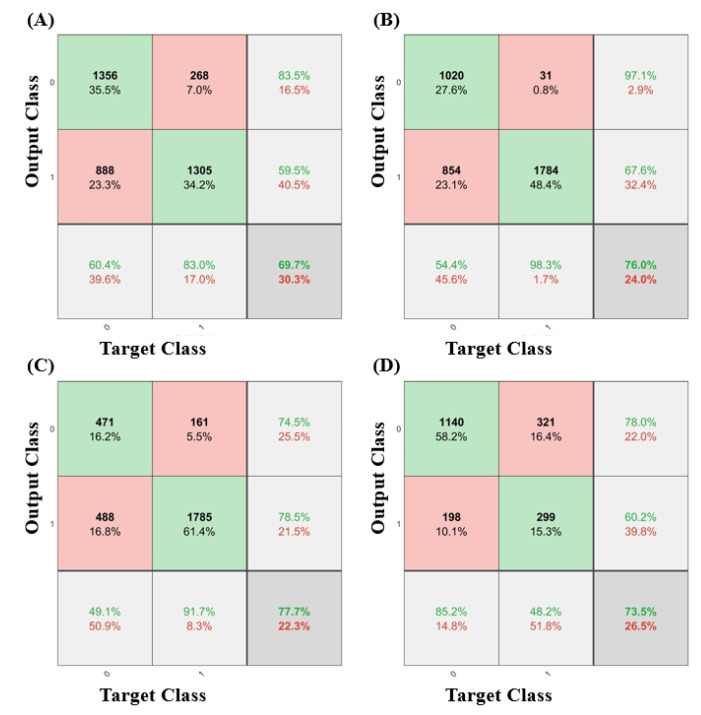
The confusion matrices of each prediction model for the validation data. Number 0 means lower ICP levels (<25 mmHg) and number 1 means IICP (≥25 mmHg): (**A**) Logistic regression; (**B**) Naïve Bayes; (**C**) Support vector machine; (**D**) Random forest.

**Table 1 diagnostics-13-00386-t001:** Baseline characteristics of study subjects according to derivation and validation group.

Study Group		Derivation Group (N = 21)	Validation Group (N = 9)	
Variables		Mean (SD)	Mean (SD)	*p*-Value
Bwt, Kg		41.3 (2.3)	44.8 (2.3)	<0.01
SBP, mmHg		108.9 (14.5)	105.3 (14.4)	<0.01
DBP, mmHg		71.2 (13.9)	65.4 (11.2)	<0.01
MAP, mmHg		83.8 (13.9)	78.7 (12.1)	<0.01
HR, beat/min		99.1 (16.9)	91.6 (11.0)	<0.01
BT, °C		37.2 (1.1)	36.2 (1.0)	<0.01
ICP, mmHg		12.4 (6.0)	15.5 (4.9)	<0.01
ABGA	pH	7.56 (0.05)	7.52 (0.03)	0.1
	pCO2	33.66 (7.10)	39.44 (3.85)	0.03
	pO2	156.92 (102.32)	113.92 (54.99)	0.25
	SpO2	98.41 (2.77)	97.96 (1.66)	0.65
	HCO3	29.84 (4.41)	32.70 (3.75)	0.1
	Hb	9.39 (1.38)	9.52 (0.89)	0.8
	Na	136.56 (4.73)	139.91 (2.65)	0.06
	LA	1.64 (0.72)	1.87 (0.49)	0.4

Bwt, Body weight; SBP, Systolic blood pressure; DBP, Diastolic blood pressure; MAP, Mean arterial pressure; HR, Heart rate; BT, Body temperature; ICP, Intracranial pressure; pCO2, Partial pressure of carbon dioxide; pO2, Partial pressure of oxygen; SpO2, Saturation pulse oxygen; HCO3, Bicarbonate; Hb, Hemoglobin; Na, Sodium; LA, Lactic acid.

**Table 2 diagnostics-13-00386-t002:** Characteristics of electroencephalography parameters in the derivation group according to increased intracranial pressure.

Variables		ICP < 20 mmHg	ICP 20~30 mmHg	ICP 30~40 mmHg	ICP 40~50 mmHg	ICP ≥ 50 mmHg	*p*-Value
		Mean (SD)	Mean (SD)	Mean (SD)	Mean (SD)	Mean (SD)	
Mean arterial pressure, mmHg		90.2 (12.2)	91.6 (19.6)	101.1 (21.6)	98.2 (24.4)	120.1 (11.2)	<0.001
Heartrate, rate/min		96.1 (16.9)	103.2 (19.6)	104.9 (20.4)	114.8 (30.4)	134.8 (52.0)	<0.001
EEG Indexes	Magnitude of EEG	32.51 (17.14)	33.54 (15.44)	37.49 (18.20)	37.68 (17.44)	41.51 (9.83)	<0.001
	DELTAR	−0.81 (0.31)	−0.72 (0.42)	−0.87 (0.40)	−0.92 (0.33)	−0.39 (0.37)	<0.001
	DTABR	0.78 (0.29)	0.70 (0.36)	0.83 (0.35)	0.86 (0.30)	0.39 (0.33)	<0.001
	THETAPR	0.14 (0.05)	0.14 (0.05)	0.12 (0.05)	0.11 (0.05)	0.08 (0.02)	<0.001
	GAMMAPR	0.02 (0.01)	0.02 (0.01)	0.02 (0.01)	0.02 (0.02)	0.07 (0.01)	<0.001
	Log energy entropy	1811.04 (475.04)	1833.23 (455.48)	1994.28 (508.89)	2008.83 (470.69)	1743.32 (282.71)	<0.001
	SD_theta	4.22 (2.56)	4.24 (2.27)	4.59 (2.62)	4.28 (2.39)	3.12 (1.18)	<0.001
	SD_alpha	2.95 (1.70)	3.14 (1.58)	3.21 (1.65)	3.06 (1.57)	3.52 (0.99)	<0.001
	SD_beta	3.19 (2.06)	3.29 (1.52)	3.41 (1.66)	3.46 (1.72)	8.10 (2.18)	<0.001
	SD_gamma	1.47 (0.90)	1.57 (0.82)	1.68 (0.95)	1.81 (0.99)	2.81 (0.63)	<0.001

ICP, intracranial pressure; SD, standard deviation; EEG, electroencephalography.

**Table 3 diagnostics-13-00386-t003:** Predictive performance of non-invasive intracranial pressure prediction models.

Models	Accuracy	Sensitivity	Specificity	Precision	F1-Score	MCC	AUC
	Mean (95% CI)	Mean (95% CI)	Mean (95% CI)	Mean (95% CI)	Mean (95% CI)	Mean (95% CI)	Mean (95% CI)
LR	0.706 (0.625–0.787)	0.535 (0.374–0.696)	0.822 (0.727–0.916)	0.649 (0.464–0.834)	0.506 (0.358–0.653)	0.322 (0.176–0.469)	0.748 (0.644–0.851)
NB	0.749 (0.658–0.840)	0.495 (0.316–0.675)	0.874 (0.760–0.989)	0.843 (0.730–0.956)	0.572 (0.427–0.717)	0.377 (0.218–0.536)	0.824 (0.740–0.907)
SVM	0.773 (0.694–0.853)	0.506 (0.315–0.697)	0.923 (0.857–0.988)	0.870 (0.755–0.985)	0.587 (0.412–0.761)	0.425 (0.259–0.591)	0.860 (0.781–0.938)
RF	0.746 (0.673–0.818)	0.654 (0.498–0.811)	0.790 (0.671–0.910)	0.670 (0.494–0.845)	0.588 (0.428–0.747)	0.403 (0.254–0.553)	0.802 (0.710–0.894)

CI, confidence interval; LR, logistic regression; NB, naïve Bayes; SVM; support vector machine; RF, random forest.

## Data Availability

Restrictions apply to the availability of these data. Data was obtained from National Research Foundation of Korea and are available from the corresponding author with the permission of National Research Foundation of Korea.

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
