# Peer review of "Prediction of Increased Intracranial Pressure in Traumatic Brain Injury Using Quantitative Electroencephalogram in a Porcine Experimental Model"

_diagnostics, 2023, doi:10.3390/diagnostics13030386_

Round 1

Reviewer 1 Report

The authors have developed an electroencephalogram model in a pig to determine if intracranial pressure (ICP) is elevated. Each of 4 different algorithms was able to determine elevated intracranial pressure with an accuracy above 0.7 with the best accuracy from the support vector machine model. The work is important, as the model could provide foundation for non-invasive ICP monitoring. The manuscript is well written.

Addressing a few comments will improve the manuscript, as follows:

1. In the ABSTRACT, the authors should keep in mind that measurement of ICP does not prevent ICP. Measurement of ICP allows recognition of ICP to facilitate institution of treatment to reduce ICP. The authors should reword the Introduction of the ABSTRACT.

2. In RESULTS, the authors should clearly state the time of EEG measurement and interpretation to determine is ICP is elevated and reduced.

3. In RESULTS, the authors should clearly state if EEG pattern changes correlate with specific ICP. This finding can be amplified in RESULTS.

Author Response

On behalf of authors, thank you for the very valuable comments by the reviewer on our paper. We have attempted to address every point commented on by the reviewer in the revised manuscript. While we believe that we have addressed all of the reviewer’s concerns, we would be more than pleased to write additional revisions if needed.

We highlighted all changes in red. Author’s answers or explanations are in blue.

Correspondent author

Ki Jeong Hong, MD, PhD.

Reviewer 2 Report

The paper presents an EEG- based Machine Learning prediction model for intracranial pressure in a swine TBI experimental model. The draft is clearly organized, conveys the information in an understandable manner, and highlights the significance of the study.

To underline the value of the paper, the authors should consider the following aspects:

1. Maybe reorganize the abstract- The authors followed the instructions in a literary manner (as in the paper), introducing Methods, Results and Conclusions as distinctively parts. I would recommend to authors, to analyses other articles from Diagnostics as example and reduce the quantitative values, just 

2. Perhaps tha authors could update the introduction section with studies related to usage of Machine Learning algorithms in prediction of intracranial pressure. 

3. The authors should better underline their results considering the importance of the study. Table 3 presents the confusion matrix of models, but I also suggest introducing a table with values predicted by the algorithms to compare the predictions values with those for derivation group. 

In addition, the authors should be more careful with the acronyms (for some acronyms, the explanation is missing) and with the required template (reference numbers should be placed in square brackets [ ], and placed before the punctuation; for example [1]).

Also fig.2 could be more meaningful by adding some indications on it.

Author Response

(The authors gave the same response as above.)
